# DSReg: Using Distant Supervision as a Regularizer

## Abstract

In this paper, we aim at tackling a general issue in NLP tasks where some of the negative examples are highly similar to the positive examples, i.e., *hard-negative examples*). We propose the *distant supervision as a regularizer (DSReg)* approach to tackle this issue. We convert the original task to a multi-task learning problem, in which we first utilize the idea of distant supervision to retrieve hard-negative examples. The obtained hard-negative examples are then used as a regularizer, and we jointly optimize the original target objective of distinguishing positive examples from negative examples along with the auxiliary task objective of distinguishing soften positive examples (comprised of positive examples and hard-negative examples) from easy-negative examples. In the neural context, this can be done by feeding the final token representations to different output layers. Using this unbelievably simple strategy, we improve the performance of a range of different NLP tasks, including text classification, sequence labeling and reading comprehension.

## 1 Introduction

Consider the following sentences in a text-classification task, in which we want to identify text describing hotels with good service/staff (as depicted as aspect-level sentiment classification in Tang et al. (2015); Li et al. (2016); Lei et al. (2016)):

- *S1: the staff are great. (positive)*
- *S2: the location is great but the staff are surly and unhelpful .. (hard-negative)*
- *S3: the staff are surly and unhelpful. (easy-negative)*

S1 is a positive example since it describes a hotel with good staff. Both S2 and S3 are negative because staff are unhelpful. However, since S2 is lexically and semantically similar with S1, standard models can be easily confused. As another example, in reading comprehension tasks like NarrativeQA Kočiský et al. (2018), truth answers are human-generated ones and might not have exact matches in the original passage. A commonly adopted strategy is to first locate similar sequences from the original passage using a pre-defined threshold (using metrics like ROUGE-L) and then treat them as positive training examples. Sequences that are semantically similar but right below this specific threshold will be treated as negative examples and will thus inevitably introduce massive labeling noise in training. This problem is ubiquitous in a wide range of NLP tasks, i.e., some of the negative examples are highly similar to positive examples. We refer to these negative examples as *hard-negative examples* for the rest of this paper. Similarly, the negative examples that are not similar to the positive examples are refered to as *easy-negative examples*. Hard-negative examples can cause big trouble in model training, because the nuance between positive examples and hard-negative examples can cause confusion for a model trained from scratch. To make things worse, when there is a class-balance problem where the number of negative examples are overwhelmingly larger than that of positive examples (which is true in many real-world use cases), the model will be at loose ends because positive features are buried in the sea of negative features.

To tackle this issue, we propose using the idea of distant supervision (e.g., Mintz et al. (2009); Riedel et al. (2010)) to regularize training. We first harvest hard-negative examples using distant supervision. This process can be done by a method as simple as using word overlapping metrics (e.g., ROUGE, BLEU or whether a sentence contains some certain keywords). With the harvested hard-negative examples, we transform the original binary classification setting to a multi-task learning setting, in which we jointly optimize the original target objective of distinguishing positive examples

from negative examples along with an auxiliary objective of distinguishing soften positive examples (comprised of positive examples and hard-negative examples) from easy-negative examples. For a neural network model, this goal can be easily achieved by using different output layers to readout the final-layer representations. In this way, the features that are shared between positive examples and hard negative examples can be captured by the model. Models can easily tell which features that can distinguish positive examples the most. Using this unbelievably simple strategy, we improve the performance of in a range of different NLP tasks, including text classification, sequence labeling and reading comprehension.

The key contributions of this work can be summarized as follows:

- We study a general situation in NLP, where a subset of the negative examples are highly similar to the positive examples. We analyze why it is a problem and how to deal with it.
- We propose a general strategy that utilize the idea of distant supervision to harvest hard-negative training examples, and transform the original task to a multi-task learning problem. The strategy is widely applicable for a variaty of tasks.
- Using this unbelievably simple strategy, we can obtain significant improvement on the tasks of text-classification, sequence-labeling and reading comprehension.

## 2 RELATED WORK

**Distant Supervision**    Mintz et al. (2009); Riedel et al. (2010); Hoffmann et al. (2011); Surdeanu et al. (2012) It is proposed to address the data sparsity issue in relation extraction. Suppose that we wish to extract sentences expressing the ISCAPITAL relation, distant supervision augments the positve training set by first aligning unlabeled text corpus with all entity pairs between which the ISCAPITAL relation holds and then treating all aligned texts as positive training examples. The idea has been extended to other domains such as sentiment analysis Go et al. (2009), computer security event Ritter et al. (2015), life event extraction Li et al. (2014) and image classification Chen and Gupta (2015). Deep leaning techniques have significantly improved the results of distant supervision for relation extraction Zeng et al. (2017); Luo et al. (2017); Lin et al. (2017); Toutanova et al. (2015).

**Multi-Task Learning (MTL)**    The idea of using data harvested via distant supervision as auxiliary supervision signals is inspired by recent progress on multi-task learning: models for auxiliary tasks share hidden states or parameters with models for the main task and act as regularizers. In addition, neural models often celebrate performance boost when jointly trained for multiple tasks Collobert et al. (2011); Chen et al. (2017); Hashimoto et al. (2017); FitzGerald et al. (2015). For instance, Luong et al. (2015) use sequence-to-sequence model to jointly train machine translation, parsing and image caption generation models. Dong et al. (2015) adopt an alternating training approach for different language pairs, i.e., they optimize each task objective for a fixed number of parameter updates (or mini-batches) before switching to a different language pair. Swayamdipta et al. (2018) propose using syntactic tasks to regularize semantic tasks like semantic role labeling. Hashimoto et al. (2017) improve universal syntactic dependency parsing using a multi-task learning approach.

## 3 MODELS

In this section, we discuss the details of the proposed model. We focus on two different types of NLP tasks, text classification and sequence labeling.

### 3.1 TEXT CLASSIFICATION

Suppose that we have text-label pairs $D = \{x_i, y_i\}$. $x_i$ consists of a sequence of tokens $x_i = \{w_{i,1}, w_{i,2}, ..., w_{i,n_i}\}$, where $n_i$ denotes the number of tokens in $x_i$. Each text $x_i$ is paired with a binary label $y_i \in \{0, 1\}$. The training set can be divided into a positive set $D^+$ and a negative set $D^-$. Let $\hat{y}_i$ denote the model prediction. The standard training objective can be given as follows:

$$
\begin{aligned}
L_1 &= - \sum_{(x_i, y_i) \in D} \log P(\hat{y}_i = y_i | x_i) \\
&= - \sum_{(x_i, y_i) \in D^+} \log P(\hat{y}_i = 1 | x_i) - \sum_{(x_i, y_i) \in D^-} \log P(\hat{y}_i = 0 | x_i)
\end{aligned}
\tag{1}
$$

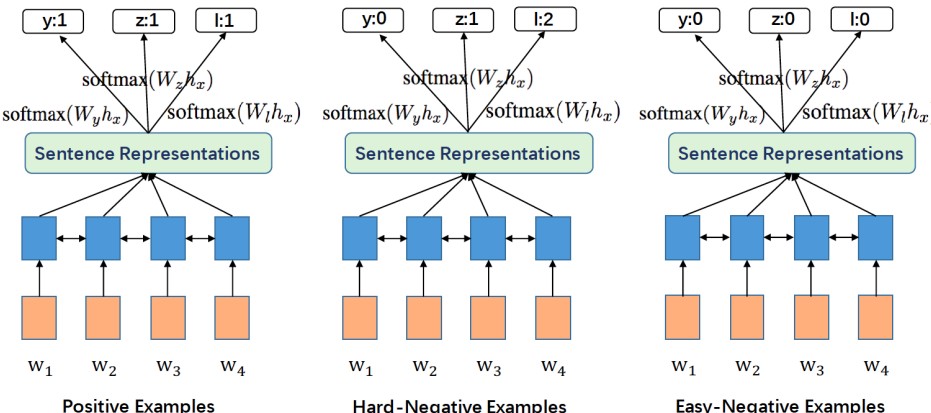

Figure 1: An illustration of the classification model with distant-supervision regularizier.

| group | notation | y | z | l |
|---|---|---|---|---|
| positive examples | $D^+$ | 1 | 1 | 1 |
| hard-negative examples | $D_{\text{hard-neg}}$ | 0 | 1 | 2 |
| easy-negative examples | $D^- - D_{\text{hard-neg}}$ | 0 | 0 | 0 |

Table 1: Labels for different instances in binary classification tasks.

Let $D_{\text{hard-neg}}$ denote the hard-negative examples retrieved using distant supervision. Here we introduce a new label $z$: $z = 1$ for instances in $D_{\text{hard-neg}} \cup D^+$, and $z = 0$ for instances in $D^- - D_{\text{hard-neg}}$. We regularize $L_1$ using an additional objective $L_2$:

$$L_2 = - \sum_{(x_i, z_i) \in D^+ \cup D_{\text{hard-neg}}} \log P(\hat{z}_i = 1 | x_i) - \sum_{(x_i, z_i) \in D^- - D_{\text{hard-neg}}} \log P(\hat{z}_i = 0 | x_i) \tag{2}$$

$L_2$ can be thought as an objective to capture the shared features in positive examples and hard-negative examples. Equ.2 can also be extended to another similar form, distinguishing between $D_{\text{hard-pos}} \cup D^-$ (i.e., the union of positive examples that are similar to negative and negative examples) and $D^+ - D_{\text{hard-pos}}$.

Empirically, we also find that adding one more three-class classification objective $L_3$, which separates positive vs hard-negative vs easy-negative introduces additional performance boost. We suggest that adding this three-class classification will additionally highlight the difference between hard negative examples and easy negative examples for the model. The label is denoted by $l$, where $l = 0$ for easy negative examples, $l = 1$ for positive examples and $l = 2$ for hard negative examples. This leads the final objective function at test time to be :

$$L = \lambda_1 L_1 + \lambda_2 L_2 + \lambda_3 L_3 \tag{3}$$

where $\lambda_1 + \lambda_2 + \lambda_3 = 1$ are used to control the relative importance of each loss. For a neural classification model, $p(z|x)$, $p(y|x)$ and $p(l|x)$ share the same model structure. The input text $x$ is first mapped to a $d$-dimensional vector representation $h_x$ using suitable contextualization strategy, such as LSTMs Hochreiter and Schmidhuber (1997), CNNs Kim (2014) or transformers Vaswani et al. (2017). Then $h_x$ is fed to three fully connected layers with softmax activation function to compute $p(y|x)$, $p(z|x)$ and $p(l|x)$ respectively:

$$\begin{aligned} p(y|x) &= \text{softmax}(W_y h_x) \\ p(z|x) &= \text{softmax}(W_z h_x) \\ p(l|x) &= \text{softmax}(W_l h_x) \end{aligned} \tag{4}$$

where $W_y, W_z \in R^{2 \times d}$, $W_l \in R^{3 \times d}$.

## 3.2 SEQUENCE LABELING

In sequence labeling tasks Lafferty et al. (2001); Ratinov and Roth (2009); Collobert et al. (2011); Huang et al. (2015); Ma and Hovy (2016); Chiu and Nichols (2016), a model is trained to assign

| group | notation | y | z | l |
|---|---|---|---|---|
| B of positive examples | $D^+$ | B | B | $B_1$ |
| I positive examples | $D^+$ | I | I | $I_1$ |
| B of hard-negative examples | $D_{\text{hard-neg}}$ | O | B | $B_2$ |
| I hard-negative examples | $D_{\text{hard-neg}}$ | O | I | $I_2$ |
| O | $D^- - D_{\text{hard-neg}}$ | O | O | O |

Table 2: Labels for different instances in sequence labeling tasks. B short for beginning and I short for inside.

labels to each of the tokens in a text sequence. Suppose that we are to assign labels to all tokens in a chunk of text $D = \{x_1, x_2, ..., x_{n_D}\}$ where $n_D$ denotes the number of tokens in $D$. Let us consider a simple case where we only have one type of tag and we will use the standard IOB (short for inside, outside, beginning) sequence labeling format. In this case, it is a three-class classification problem, assigning $y_i \in (B, I, O)$ to each token. We treat tokens with label $B$ and $I$ as $D^+$ and tokens with label $O$ as $D^-$. The objective function for the vanilla sequence labeling task is given as follows:

$$L_1 = -\log P(y_{1:n_D}|x_{1:n_D}) \tag{5}$$

$P(y_{1:n_D}|x_{1:n_D})$ can be computed using standard sequence tagging models such as CRF Lafferty et al. (2001), hybrid CRF+neural models Huang et al. (2015); Ma and Hovy (2016); Chiu and Nichols (2016); Ye and Ling (2018) or purely neural models Collobert et al. (2011); Devlin et al. (2018).

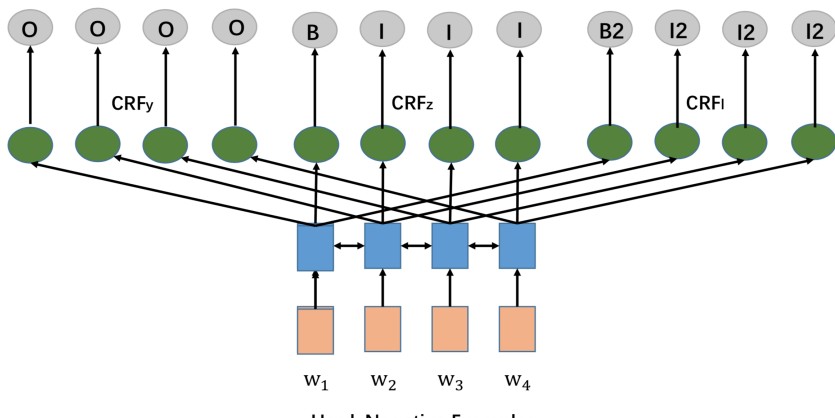

Figure 2: An illustration of the CRF tagging model with the distant-supervision regularizer.

To take into account negative examples that are highly similar to positive ones, we use the idea of distant supervision to first retrieve the set of hard-negative dataset $D_{\text{hard-neg}}$. Akin to the text classification task, we introduce a new label $z_i \in (B, I, O)$, indicating whether the current token belongs to $D_{\text{hard-neg}}$. To incorporate the collected hard-negative examples into the model, again we introduce an auxiliary objective of assigning correct $z$ labels to different tokens:

$$L_2 = -\log P(z_{1:n_D}|x_{1:n_D}) \tag{6}$$

Similar to the text classification task, we also want to separate hard negative examples, easy negative examples and positive examples, so we associate each example with a label $l$. We thus have distinct "outside" and "beginning" labels for positive examples and hard-negative examples, i.e., $l \in (B_{\text{pos}}, I_{\text{pos}}, B_{\text{hard-neg}}, I_{\text{hard-neg}}, O)$. Labels for different categories regarding $y$ and $z$ are shown in Table 2. The final objective function is thus as follows:

$$L = \lambda_1 L_1 + \lambda_2 L_2 + \lambda_3 L_3 \tag{7}$$

Again $\lambda_1 + \lambda_2 + \lambda_3 = 1$. At the training time, the two functions are simultaneously trained. At the test time, we only use $P(y_i|x_{1:n_D})$ to predict $y_i$ as the final decision.

For CRF-based models Huang et al. (2015); Ma and Hovy (2016); Chiu and Nichols (2016), neural representations are fed to the CRF layer and used as features for decision making. As in Ma and Hovy (2016), neural representation $h_{x_i}$ is computed for each token/position using LSTMs and CNNs, and then forwarded to the CRF layer. The key issue with CRF-based models is that the CRF model is only

able to output one single label. This rules out the possibility of directly feeding $h_x$ to three readout layers to simultaneously predict $y$, $z$ and $l$. We propose the following solution: we use three separate CRFs to predict $y$ (for $L_1$), $z$ (for $L_2$) and $l$ (for $L_3$), denoted by $CRF_y$, $CRF_z$ and $CRF_l$. The three CRFs use the same hidden representation $h_x$ obtained from the same neural model as inputs, but independently learn their own features weights. We iteratively perform gradient descent on the three $CRF$s, and the error from the three $CRF$s are back-propagated to the neural model iteratively.

## 4 EXPERIMENTS

### 4.1 TEXT CLASSIFICATION

For the text classification task, we used three datasets, the Stanford Sentiment Treebank dataset (SST) Socher et al. (2013), the hotel review dataset for aspect-specific sentiment analysis in Lei et al. (2016) and the financial statement dataset that we created.

**The Stanford Sentiment Treebank (SST) Dataset** associates each phrase within a sentence with a sentiment label. The task is originally further divided into two-class coarse-grained classification and five-class fine-grained classification. We only report binary classification results. For baseline, we used BERT-large Devlin et al. (2018). The representation for CLS is output to the softmax layer for classification. We used distant supervision to retrieve hard-negative examples: similar to Mintz et al. (2009), we employed a simple strategy, in which we treat negative reviews that contain positive sentiment lexicons as hard-negative examples. Positive sentiment words are retrieved using the MPQA corpus Wilson et al. (2005).

**The Hotel Review Dataset** contains roughly 50,000 reviews with an average length of 120 words. Each review contains ranking scores (integers from 1 to 5) for different aspects of the hotel, such as service, cleanliness, location, rooms, etc. The dataset is constructed in a way that each review might contain diverse sentiments towards different aspects, and it is interesting to see how a model manages or fails to identify these different aspects and their associated scores when entangled with other aspects. Following Li et al. (2016), the task is divided into four sub-tasks, each of which classifies the sentiment of one of the following four aspects: we focus on four aspects: value, rooms, service and location. We filtered neutral reviews (with score 3) and treat those with score 1 and 2 as negative ones, 4 and 5 as positive one. The task is thus transformed to a binary classification task. The same review can thus carries positive sentiment regarding one aspect, but negative for the other. We report average accuracy for the four tasks. For fair comparison, we used the baseline in Li et al. (2016). The model combines Bi-LSTMs with a memory-network structure Sukhbaatar et al. (2015) at both the word level and the sentence level to obtain document-level representations, which are then fed to a sigmoid function for binary classification. Hard-negative examples are retrieved is the same way as in the SST dataset.

**The Financial Statement Dataset** is a dataset created by us. It consists 97,736 sentences, each of which contains labels for the values of financial statement items (FSI) for individual businesses or services from annual reports of listed companies (Statistics shown in Table 6). The goal is to extract:

- the [$value$] of [$which\ financial\ statement\ item$] of [$which\ business\ or\ service$] of a listed company

The dataset can be used for both text classification and sequence labeling. For the task of text classification, the goal to identify whether a sentence contains useful FSI information, i.e., whether a sentence contains the FSI of interest and the corresponding value. Sentences with annotated financial statement items and the corresponding values as positive examples, and the rest are negative examples.

We used the following distant-supervision pattern to retrieve hard-negative examples: we treat texts in negative examples that contain more than one mention of financial statement items as hard-negative examples. For example, the sentence *Benefiting from the decline in raw material costs and the further improvement of the company's management skills, the profit of the company increased during the reporting period* is a hard-negative example. Particularly, it is a negative example since it does not specifically indicate the value of the profit and thus is not of interest. But this sentence is a hard-negative one because it contains the FSI keyword *profit*.

| Model | SST | Hotel Review | Financial Statement |
|---|---|---|---|
| $L_1$ | 81.5 | 86.2 | 88.4 |
| Pos∪HardNeg\|EasyNeg → Pos\|HardNeg | 82.4 | 86.9 | 88.0 |
| $L_3$ | 79.2 | 85.8 | 81.3 |
| $L_1 + L_2$ | 82.7 | 87.9 | 89.4 |
| $L_1 + L_3$ | 82.4 | 87.1 | 88.5 |
| $L_1 + L_2 + L_3$ | **82.9** | **88.0** | **90.8** |

Table 3: Performances of different models on the text classification task.

| Bi-LSTM + CRF | | | |
|---|---|---|---|
| Model | P | R | F |
| $L_1$ | 82.06 | 83.26 | 82.65 |
| $L_1 + L_2$ | 82.41 | **85.51** | 83.93 |
| $L_1 + L_3$ | 82.22 | 85.14 | 83.65 |
| $L_1 + L_2 + L_3$ | **83.92** | 84.97 | **84.44** |

Table 4: Performances of different models on the sequence labeling task on the Financial Statement Dataset.

For the ease of notations, we use Pos and Neg to denote positive and negative examples, HardNeg to denote hard-negative examples and EasyNeg to denote easy-negative examples. We compare performances of the following models:

- $L_1$: the vanilla classification model to distinguish between positive examples (Pos) and negative examples.
- Pos∪HardNeg\|EasyNeg → Pos\|HardNeg: a hierarchical model that involves two states: 1) distinguishing between easy-negative (EasyNeg) and the union of positive and hard-negative (Pos∪HardNeg) 2) distinguishing positive (Pos) from hard-negative (HardNeg) examples.
- $L_3$: a three-class classification model to distinguish Pos, HardNeg and EasyNeg.
- $L_1 + L_2$: the proposed multi-task learning model that jointly trains two objective functions: distinguishing Pos from Neg and positive+easy negative (Pos∪HardNeg) from hard negative (EasyNeg) examples.
- $L_1 + L_3$: combining the standard classification objective with the three class classification objective.
- $L_1 + L_2 + L_3$: combining the three.

**Results** Table 3 shows results for text classification tasks. As can be seen, three versions of the proposed DSReg models, i.e., $L_1+L_2$, $L_1+L_3$ and $L_1+L_2+L_3$, outperform the binary classification model and the pipelined model, which aligns with our expectation. For the pipelined model, since the error accumulates over stages, it underperforms not only the DSReg models, but also the binary classification model. By using both the three-class classification ($L_3$) and Pos∪HardNeg\|EasyNeg ($L_2$) as regulations, the $L_1 + L_2 + L_3$ setting leads to the best performance.

## 4.2 SEQUENCE LABELING

For the sequence labeling task, again we used the Financial Statement Dataset. The task can be transformed to assigning IOB (short for inside, outside, beginning) of each label category to each word. Labels to extract include *financial statement items* (FSI), *unit*, *value*, *the change of FSI* (FSI-change), *time*, *business & service* (B&S), *bases of comparison* (BoC), as illustrated in the following example:
*The (O) revenue (B-FSI) of education (B-B&S) sector (I-B&S) increased (B-FSI-change) by 25% (B-value) over (O) the (B-BoC) same (I-BoC) period (I-BoC-I) of (I-BoC) the (I-BoC) previous (I-BoC) year (I-BoC) . (O)*

We used the keyword matching strategy to retrieve hard-negative examples. We report statistics at the word-level in Table 4. The three-class classification model ($L_3$) Pos\|HardNeg\|EasyNeg and the pipelined model Pos∪HardNeg\|EasyNeg → Pos\|HardNeg significantly underperform the others, and their results are omitted due to space limitations. From Table 4, we can see that the proposed $L_1 + L_2 + L_3$ model significantly outperforms the Pos\|Neg baseline by (+1.79).

### 4.3 Reading Comprehension

| Model | Standard $L_1$ | DSReg $(L_1 + L_2)$ | DSReg $(L_1 + L_2 + L_3)$ | Human |
|---|---|---|---|---|
| | | summary: valid/test | | |
| BLEU-1 | 33.45/33.72 | 34.89/34.90 | 35.12/35.02 | 44.24/44.43 |
| BLEU-4 | 15.69/15.53 | 16.89/16.89 | 17.17/17.14 | 18.17/19.65 |
| Meteor | 15.68/15.38 | 17.05/16.72 | 17.21/16.84 | 23.87/24.14 |
| ROUGE-L | 36.74/36.30 | 38.40/37.65 | 38.55/37.90 | 57.17/57.02 |
| Model | Standard | DSReg $(L_1 + L_2)$ | DSReg $(L_1 + L_2 + L_3)$ | Human |
| | | full version: valid/test | | |
| BLEU-1 | 5.82/5.68 | 7.60/7.55 | 7.81/7.77 | 44.24/44.43 |
| BLEU-4 | 0.22/0.25 | 0.35/0.41 | 0.37/0.37 | 18.17/19.65 |
| Meteor | 3.84/3.72 | 5.17/5.02 | 5.22/5.05 | 23.87/24.14 |
| ROUGE-L | 6.33/6.22 | 7.40/7.17 | 7.66/7.22 | 57.17/57.02 |

Table 5: Results on the NarrativeQA dataset.

The narrativeQA dataset Kočiskỳ et al. (2018) consists of 1,567 stories with 46,765 question-answer pairs. Following Kočiskỳ et al. (2018), we conduct experiments on both the summary setting and the full version setting. For the summary setting, answer spans need to be extracted from story summaries, and for the full version setting, they need to be extracted from the entire books or movie scripts. We followed the routines of the neural benchmarks in Kočiskỳ et al. (2018), in which we first retrieve relevant chunks from the story using an IR system. Then we concatenate the selected chunks. Since the answer for each query is manually annotated, a large proportion of answers do not have corresponding spans in the original passage that can be exactly matched. In Kočiskỳ et al. (2018), the span that achieves the highest ROUGE-L score with respect to the reference answer are used as gold spans. The start and end indices are predicted using BiDAF Seo et al. (2016). As in Kočiskỳ et al. (2018), spans with highest ROUGE-L scores are treated as positive examples. Suppose that for a gold answer $a$, text span $a'$ has the highest ROUGE-L score of $\text{ROUGE}(a, a')$, and is thus treated as the positive training example. Since documents and passages in this task are very similar, there are many text spans that are highly similar to the gold answer. These spans are treated as hard-negative examples. Specifically, we treat spans whose ROUGE-L scores are greater than $\alpha \times \text{ROUGE}(a, a')$ as hard-negative examples, where $\alpha \in (0, 1)$ is the parameter to tune on the development set.

We followed the criteria in Kočiskỳ et al. (2018) to train a vanilla BiDAF model. We used the splits of positive examples, hard-negative examples and easy-negative examples to train DSReg models, using both the three-class classification and the (positive+hard-neg) vs. easy-neg as regulations. We report scores for BLEU-1, BLEU-4 Papineni et al. (2002), Meteor Denkowski and Lavie (2011) and ROUGE-L scores Lin (2004). Results are shown in Table 5. When combined with BiDAF, the proposed DSReg model outperforms the standard BiDAF model in both settings. As discussed in Kočiskỳ et al. (2018), the span prediction model performs the best in the summary setting. This sets new state-of-the-art results for the summary setting on the NarrativeQA dataset.

## 5 Ablation Study and Visualization

### 5.1 The Effect of $L_2$: Pos∪HardNeg|EasyNeg

We examine the influence of the parameter $\lambda$ at the interval of 0.1 in the objective $L = (1-\lambda)L_1 + \lambda L_2$ on the hotel review dataset. It is worth noticing that $\lambda$ cannot take the value of 1 here, since we are not able to make decisions purely based on $L_2$. Results are shown in Figure 3. The performance first goes up when $\lambda$ is smaller than 0.4, and then declines dramatically. It accords with our expectation that these losses are complementary to each other.

### 5.2 The Effect of $L_3$: Pos|HardNeg|EasyNeg

We examine the influence of the parameter $\lambda$ at the interval of 0.1 in the objective $L = (1-\lambda)L_1 + \lambda L_3$ on the hotel review dataset. Results are shown in Figure 3. Best performance is obtained when $\lambda$ is set to 0.3.

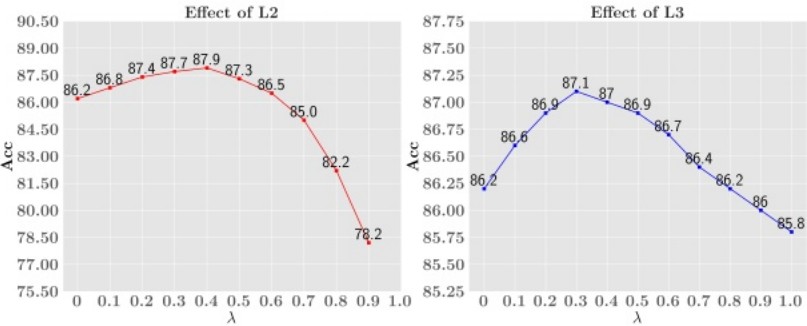

Figure 3: Effect of $L_2$ and $L_3$.

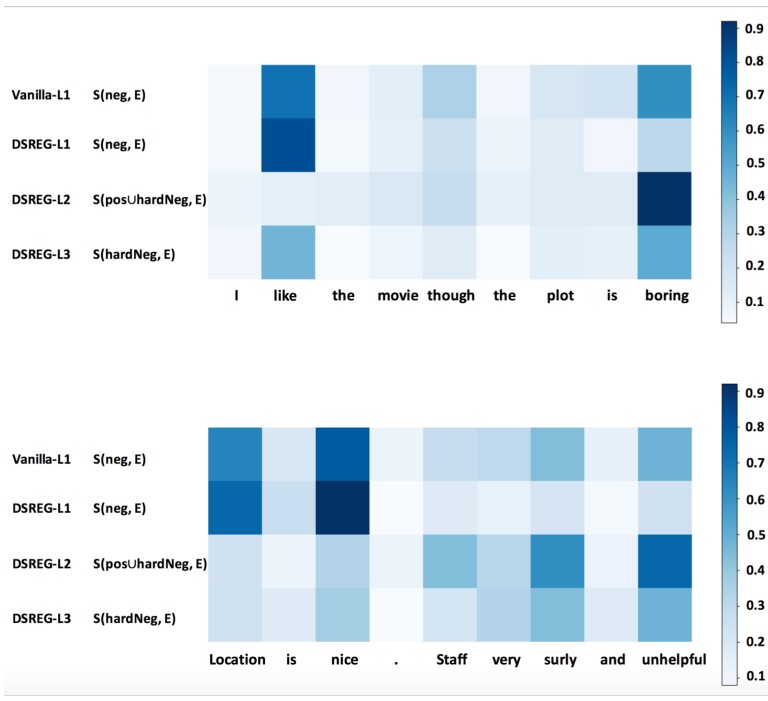

Figure 4: Saliency maps from different models.

## 5.3 VISUALIZATION

For visualization, we use FIRST DERIVATIVE SALIENCY to unveil why DSReg helps. First Derivative
Saliency is a commonly adopted visualization technique to see how much each input unit contributes
to the final decision using first-order derivatives Erhan et al. (2009); Li et al. (2015); Simonyan et al.
(2013). Given word embedding $E = \{e\}$, where $e$ denotes the value of each embedding dimension,
and a class label $c$, the trained model associates the pair $(e, c)$ with a log likelihood $S(c, e)$. For neural
models, the class score $S(c, e)$ is a highly non-linear function. $S(c, e)$ is approximated with a linear
function of $e$ by computing the first-order Taylor expansion

$$S(c, e) \approx w(c, e)^T e + b \tag{8}$$

where $w(c, e)$ is the derivative of $S(c, e)$ with respect to the embedding dimension $e$.

$$w(c, e) = \frac{\partial S(c, e)}{\partial e} \Big|_e \tag{9}$$

The magnitude (absolute value) of the derivative indicates the sensitiveness of the final decision to
the change in one particular dimension, telling us how much one specific dimension of the word

embedding contributes to making the decision $c$. The saliency score of a specific word $E$ is the average of the absolute value of $w(c, e)$:

$$S(c, E) = \frac{1}{|E|} \sum_{e \in E} |w(c, e)| \qquad (10)$$

We use the example "I hate the movie though the plot is interesting" in SST and "Location is nice. Staff very surly and un helpful" in aspect sentiment identification to show the working mechanism of different constituents in DSREG to illustrate why it works better. Both examples consist of two clauses with opposite sentiments, which inevitably makes a model confused.

Figure 4 shows saliency maps regarding each word output from different models. Vanilla denotes the baseline classification model, the objective of which consists of only $L_1$. As can be seen from S(pos, E) of Vanilla-L1 , though the model emphasizes more on hate in the first clause, but attaches significant amount of importance to "interesting" in the second clause. This issue is largely alleviated in S(pos, E) of DSREG-L1, which offers a clearer focus on the first clause than the second one. The reason why this happens can be explained by the saliency map from S(pos∪hardNeg, E) of DSREG-L1: the example is hard-negative (it is negative but contains positive lexicon "interesting") and the sentiment from the distracting word "interesting" is captured in the S(pos∪hardNeg, E), making S(pos, E) more easily focus on the truly positive clause.

## 6 CONCLUSION

In this paper, we tackle a general problem in NLP, i.e., the situation in which a subset of the negative examples are highly similar to the positive ones. We propose the DSReg: a model that utilizes distant supervision as a regularizer. We transform the original task to a multi-task learning problem, in which we first utilize distant supervision to retrieve hard-negative examples, which are then used as a regularizer. We show that the proposed strategy lead to significant performance boost text classification, sequence labeling and machine reading comprehension tasks.

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

## 7 APPENDIX

| Tag | examples | # of instances |
|---|---|---|
| financial statement items (FSI) | income; revenue | 6543 |
| unit | ton; Kilowatt hour | 5622 |
| value | 5;100,000; 25% | 9073 |
| bases of comparison (BoC) | the same period of last year | 2585 |
| the change of FSI (FSI-change) | increase; decrease | 2821 |
| business & service (B&C) | education sector; loan service | 1617 |

Table 6: Details for labels of the financial statement item dataset.

