# OpenReview forum: "DSReg: Using Distant Supervision as a Regularizer"
_ICLR.cc/2020/Conference — Reject_

### Official Review · AnonReviewer1 · 2019-10-22
**Official Blind Review #1**

**Rating:** 3

**Review:**

This paper proposes to improve performance of NLP tasks by focusing on negative examples that are similar to positive examples (e.g. hard negatives). This is achieved by regularizing the model using extra output classifiers trained to classify examples into up to three classes: positive, negative-easy, and negative-hard. Since those labels are not provided in the original data, examples are classified using heuristics (e.g. negative examples that contain a lot of features predictive of a positive class will be considered as hard-negative examples), which are used to provide distant supervision. This general approach is evaluated on phrase classification tasks, one information extraction task, and one MRCQA task.

Although the proposed approach is interesting, this paper has several weaknesses (i) the method is not sufficiently justified or analyzed; (ii) there are missing links with previous work (notably on domain adversarial training); (iii) experimental setting is rather weak.

1) About the justification of the approach:
1.1) I feel like the proposed are not intuitively justified enough. They point out that "L_2 can be thought as an objective to capture the shared features in positive examples and hard-negative examples". Why would that be good from an intuitive perspective ?
1.2) L_3 is forcing the model to group all the hard-negative examples together. Do you have an intuition why would that be useful ?
1.3) What happens if the model overfits the hard negative examples in the training set ? This would mean that it has captured some features that can distinguish positive / negative label. Why would L_3 help in that case ?

2) About related approaches:
2.1) How does this method relate to domain adversarial training applied to positive and hard-negatives and adversarial examples in general ?
2.2) Would similar performance be obtained by virtual adversarial training for example ?

3) About the experimental setting:
3.1) The performance reported is well below the use of recent work on these datasets and recent models such as BERT. Would these improvements carry over to bigger architectures ?
3.2) In SST, the paper say they use BERT large, but the baseline performance (81.5) is well below BERT large performance in the original paper (94.9, https://arxiv.org/pdf/1810.04805.pdf). Why the mismatch ?
3.3) What's the proportion of hard-negative examples mined for training and test set ? While the heuristics used seem reasonable, without those numbers, it is impossible to know if the heuristics truly predict hard-negative examples.
3.4) Does the performance gain comes from better predicting hard-negative examples in the test set ? One could analyze the performance per error type (i.e. true positive, false negative, false positive (easy), false positive (hard)) with the baseline model and of the various proposed regularizing tasks (e.g. L_2 and L_3, both in training and test).
3.5) The heuristics are used to pick what should be adversarial examples, but there is no mentions of this concept in the text. Oversampling those adversarial examples could, potentially, improve the performance of the baseline model. It would be interesting to try this.
3.6) If possible, it would be good to add standard deviation to the results obtained running multiple runs.
3.7) The visualization sub-section is anecdotal and not especially illuminating, and its text seems to refer to a different example than the figures ("interesting" is not in the figures.)

Minor points:

- Section 3 (Models) may be made shorter, the models used are utterly simple. This could free up space for more experiments.
- In the tables, simply adding the name of the used model to the "L1" rows would be clearer.
- The description of the pipelined results in Section 4.1 does not match the results shown in the table.
- The citations are not well integrated with the text (\citep vs \cite), and the formatting of CRF changes in the last paragraph of Section 3.2.


**Experience Assessment:**

I have published one or two papers in this area.

**Review Assessment: Checking Correctness Of Derivations And Theory:**

I assessed the sensibility of the derivations and theory.

**Review Assessment: Checking Correctness Of Experiments:**

I assessed the sensibility of the experiments.

**Review Assessment: Thoroughness In Paper Reading:**

I read the paper at least twice and used my best judgement in assessing the paper.

---

> ### Author Response · Authors · 2019-11-09
> **response to Review #1**
>
> About the justification of the approach:
> 1.1) I feel like the proposed are not intuitively justified enough. They point out that "L_2 can be thought as an objective to capture the shared features in positive examples and hard-negative examples". Why would that be good from an intuitive perspective ?
> 1.2) L_3 is forcing the model to group all the hard-negative examples together. Do you have an intuition why would that be useful ?
> 1.3) What happens if the model overfits the hard negative examples in the training set ? This would mean that it has captured some features that can distinguish positive / negative label. Why would L_3 help in that case ?
>
> Regarding the intuition:
> Sorry for the confusion. Let me use an analogy to describe the intuition behind what this paper is doing. Think about a situation where we want to separate eagles (positive examples) from animals that are not eagles (negative examples). The tricky part here is that we have birds-that-are-not-eagles. Since they are not eagles, they are negative examples. But they do share a lot of common features with eagles, making the model hard to distinguish them from eagles.  birds-that-are-not-eagles are hard-neg examples in this paper. The rest, animals-that-are-not-birds are easy-neg examples.
>
> re: They point out that "L_2 can be thought as an objective to capture the shared features in positive examples and hard-negative examples". Why would that be good from an intuitive perspective ?
> L2 separates pos+hard-neg from easy-neg. In our analogy, its function is to separate  birds (eagles+ birds-that-are-not-eagles= birds) from animals-that-are-not-birds.  Then "Why would that be good from an intuitive perspective"? Since L2 learns the birds features, which are shared by both eagles and birds-that-are-not-eagles, it would be then easier for models to learn what are not shared between them using L1 and L3, leading to better performance.
>
> re: 1.2) L_3 is forcing the model to group all the hard-negative examples together. Do you have an intuition why would that be useful ?
> Why is  grouping hard-negative examples useful? We are grouping animals-that-are-not-birds and training a classifier to separate eagles vs birds-that-are-not-eagles vs animals-that-are-not-birds. It is intuitive that this might help.

---

### Official Review · AnonReviewer2 · 2019-10-23
**Official Blind Review #2**

**Rating:** 6

**Review:**

The authors propose a novel approach to leverage Distant Supervision for discriminating between positive examples and "negative examples that share salient features with the positive ones." In spite of its simplicity, the method appears to be quite promising.

The main feedback for the authors is to describe "early & in detail" the distant supervision techniques used in the experiments. The paper would be greatly improved by adding:
- an intuitive paragraph in the intro that explains a concrete example of DS high level, but with enough details for the reader to grasp the idea)
- adding a new section right after related work (and before the current "3. Models") in which you present in great detail (and with concrete & complete examples) the two main DS techniques that are used in the experiments; with that solid understanding


Other comments:
- for sake of simplicity & understand-ability, you should avoid the use of the term "soften positive examples"  in the abstract
- avoid using the term "unbelievable" (one in abstract & twice in intro"
- the last paragraph before "Conclusion" seems to refer to an earlier version of Figure 4, which, in its current form, does NOT
have S(pos,E) or the word "interesting"

**Experience Assessment:**

I have read many papers in this area.

**Review Assessment: Checking Correctness Of Derivations And Theory:**

I assessed the sensibility of the derivations and theory.

**Review Assessment: Checking Correctness Of Experiments:**

I assessed the sensibility of the experiments.

**Review Assessment: Thoroughness In Paper Reading:**

I read the paper at least twice and used my best judgement in assessing the paper.

---

### Official Review · AnonReviewer3 · 2019-10-23
**Official Blind Review #3**

**Rating:** 3

**Review:**

This paper is aimed at tackling a general issue in NLP: Hard-negative training data (negative but very similar to positive) can easily confuse standard NLP model. To solve this problem, the authors first applied distant supervision technique to harvest hard-negative training examples and then transform the original task to a multi-task learning problem by splitting the original labels to positive, hard-negative, and easy-negative examples. The authors consider using 3 different objective functions: L1, the original cross entropy loss; L2, capturing the shared features in positive and hard-negative examples as regularizer of L1 by introducing a new label z; L3, a three-class classification objective using softmax.
This authors evaluted their approach on two tasks: Text Classification and Sequence Labeling. This implementation showed improvement of performance on both tasks.

Strenghts:
+ the paper proposes a reasonable way to try to improve accuracy by identifying hard-negative examples
+ the paper is well written, but it would benefit from another round of proofreading for grammar and clarity

Weaknesses:
- performance of the proposed method highly depends on labels of hard-negative examples. The paper lacks insight about a principled way to label such examples, the costs associated with such labeling, and impacts of the labeling quality on accuracy. The experiments are not making a convincing case that similar improvements could be obtained on a larger class of problems.
- The objective function L3 is not well justified.
- It would be important to see if the proposed method is also beneficial with the state of the art neural networks on the two applications.
- Table 3 (text classification result) does not list baselines.

**Experience Assessment:**

I have published one or two papers in this area.

**Review Assessment: Checking Correctness Of Derivations And Theory:**

I assessed the sensibility of the derivations and theory.

**Review Assessment: Checking Correctness Of Experiments:**

I assessed the sensibility of the experiments.

**Review Assessment: Thoroughness In Paper Reading:**

I read the paper at least twice and used my best judgement in assessing the paper.

---

> ### Author Response · Authors · 2019-11-09
> **response to review#3**
>
> thank you for the sensible comments.
>
> re: performance of the proposed method highly depends on labels of hard-negative examples . The paper lacks insight about a principled way to label such examples,
>
> We really appreciate your sensible and sharp comment. We completely agree with the point that there is no unified way to harvest hard-negative examples. But I think this does not hinder the contribution of this paper:
>
> First, just as the Mintz et al.,09 's paper shows that it is useful to use the idea of distant supervision to harvest/augment training data, we think it is similarly meaningful to show that you can use the distant supervision idea identify hard-negative examples, and your model will improve.
>
> Second, the fact that there is no unified and principled way to harvest negative examples is not because the flaws of the proposed model, but because different tasks are just different, and definitions for hard-negative examples are intrinsically different for different tasks.
> If we look at the distant supervision literature,  we  can find that how people use the idea of distant supervision to help different tasks are very different, e.g., for relation extraction, researchers use triples in freebase to harvest sentences containing the mention of these triples (Mintz et al.,2009  Distant supervision for relation extraction without labeled data); for sentiment analysis, researchers use emoticon to harvest training data ( Go et al., 2009  Twitter sentiment classification using distant supervision); in security researchers use events in calendar to harvest training data.  Using task-specific data harvesting methods do not prevent them from being very influential (and highly cited) papers.
>
>
> re: - The objective function L3 is not well justified.
> Thank you for your comment. We will make this point clear and justified. L3 (pos vs easy-neg vs hard-neg) is of the same nature with the combination of L1 (pos vs  easy-neg + hard-neg) and L2 (pos + hard-neg vs easy-neg). L1+L2 can actually fully express L3:  using L1 we can know which examples are positive,  using L2 we can know which examples are easy-neg, and the rest are hard-neg.
> Adding L3 can empirically help the model, since it makes it very explicit that easy-neg and hard-neg are different.
>
>
> re: It would be important to see if the proposed method is also beneficial with the state of the art neural networks on the two applications.
> Thank you for the comment.  For text classification, we use  BERT_large as a baseline. For sequence labeling, we use BERT_large+CRF as a baseline. For these two tasks, we believe we did adopt SOTA (or nearly SOTA) neural structures.
> For reading comprehension, BiDAF was used, which is not SOTA. We redid experiments and used SpanBERT. The proposed model still consistently outperforms baselines when using SpanBERT as a backbone:
>
>                 L1(baseline)           L1+L2          L1+L2+L3          Human
> BLEU-1     35.20/35.33    36.59/36.71    37.04/37.11     44.24/44.43
> BLEU-4     16.29/16.41    17.08/17.11    17.32/17.40     18.17/19.65
> Meteor     16.90/16.65    18.32/18.38    18.77/18.92     23.87/24.14
> ROUGE-L  39.89/39.78    41.43/41.38    42.21/42.17     57.17/57.02
>
> re: - Table 3 (text classification result) does not list baselines.
> Sorry for the confusion. Actually, the first line L1 is the baseline, since L1 denotes the objective only based on original golden labels  (see Eq1 and Eq5 ). We will make this point more clear in the updated version.

---

### Decision · Program_Chairs · 2019-12-19

**Decision:**

Reject

**Comment:**

This paper proposes a way to handle the hard-negative examples (those very close to positive ones) in  NLP, using a distant supervision approach that serves as a regularization.   The paper addresses an important issue and is well written; however, reviewers pointed put several concerns, including testing the approach on the state-of-art neural nets, and making experiments more convincing by testing on larger problems.